Matching based on biological categories in Orangutans (Pongo abelii) and a Gorilla (Gorilla gorilla gorilla)

Vonk Jennifer vonk@oakland.edu
Department of Psychology, Oakland University , Rochester, MI , United States
Rogers Lesley
Electronic publication date: 2013 Sep 10
Publication date: 2013
Volume: 1
Electronic Location ID: e158
Received 2013 Jul 22; Accepted 2013 Aug 21
Copyright: © 2013 Vonk
Copyright year: 2013
Copyright holder: Vonk
License: This is an open access article distributed under the terms of the Creative Commons Attribution License, which permits unrestricted use, distribution, and reproduction in any medium, provided the original author and source are credited.
License URL: https://creativecommons.org/licenses/by/3.0/

Keywords: Gorilla, Matching, Orangutans, Biological categories, Concepts

Funding: Natural Sciences and Engineering Research Council (NSERC) This research was supported by a Natural Sciences and Engineering Research Council (NSERC) Postgraduate scholarship granted to the author, and an NSERC Operating grant to Suzanne E. MacDonald. The funders had no role in study design, data collection and analysis, decision to publish, or preparation of the manuscript.

==============================
Following a series of experiments in which six orangutans and one gorilla discriminated photographs of different animal species in a two-choice touch screen procedure, Vonk & MacDonald (2002) and Vonk & MacDonald (2004) concluded that orangutans, but not the gorilla, seemed to learn intermediate level category discriminations, such as primates versus non-primates, more rapidly than they learned concrete level discriminations, such as orangutans versus humans. In the current experiments, four of the same orangutans and the gorilla were presented with delayed matching-to-sample tasks in which they were rewarded for matching photos of different members of the same primate species; golden lion tamarins, Japanese macaques, and proboscis monkeys, or family; gibbons, lemurs (Experiment 1), and subsequently for matching photos of different species within the following classes: birds, reptiles, insects, mammals, and fish (Experiment 2). Members of both Great Ape species were rapidly able to match the photos at levels above chance. Orangutans matched images from both category levels spontaneously whereas the gorilla showed effects of learning to match intermediate level categories. The results show that biological knowledge is not necessary to form natural categories at both concrete and intermediate levels.

Introduction

Rosch et al. (1976) suggested a chronological and hierarchical structure for concepts, such that humans first learn basic level concepts (e.g., dog) and only later are able to learn subordinate (e.g., poodle) and superordinate (e.g., mammal or animal) concepts (see also Mervis & Rosch, 1981). This chronology of concept learning seems tied to language in that the basic level category words are the first learned and the most commonly used. It has been suggested that the formation of superordinate categories relies less upon perceptual feature analysis and more on an understanding of how the category coheres, across significant perceptual variance (Spalding & Ross, 2000). Superordinate categories are thus thought of as being more conceptually based or “abstract” relative to basic level categories, and it has been speculated that language is necessary for the formation of these later categories (Benelli, 1988; Keil, 1988; Nelson, 1988; Premack, 1983). Organisms may be inherently preprogrammed to distinguish between items at the level of the basic concept perceptually (Cerella, 1979; Eimas & Quinn, 1994). However, Mandler (2000) has articulated an opposing view, supported by several studies, in which the conceptualization of superordinate categories by human infants may actually precede that of basic level categories. Mandler’s description highlights the distinction between perceptual and conceptual categorization, a distinction not traditionally acknowledged by other researchers. In her view, exemplars within basic level categories may be associated together on a perceptual basis before exemplars from superordinate categories are associated perceptually. In contrast, when forming conceptual categories, which are based on shared, underlying properties as opposed to perceptual similarity, more global, abstract categories such as animals, foods, etc., may emerge first. This may be the case because children learn about broad categories, such as animate/inanimate distinctions prior to learning specific distinctions such as between reptiles and mammals.

Recent work (Coley, 2007) has indicated that children may make global distinctions such as between animals and non-animals, but are less likely, compared to adults, to consider humans to be similar to other primates or non-primate animals. This recent work, highlighting differences between children and adults with regard to between categorical judgments of similarity, reinforces the notion that such categorization is not made on a purely perceptual basis. In addition, Gelman & Davidson (2013) have shown that both children and adults use category membership more than similarity to make basic level category inferences, although their studies involved experimenter-created rather than actual natural categories. Thus there is some disagreement as to what sorts of categorization depend upon physical similarity, depend upon language and emerge first in the human infant. By studying concept formation in non-human primates one can determine parallels in the emergence of non-language based categorization. In the current study, two species of great ape (orangutans and a gorilla) were required to match images based on biological classifications at the level of species, family or class.

Although taxonomic class groupings may be considered basic level concepts (Roberts & Mazmanian, 1988), it has been suggested that the ability to make taxonomic classifications of natural stimuli depends upon biological or scientific knowledge (Eimas & Quinn, 1994; Hampton, 1998; Inagaki, 1989), as well as language (Anggoro, Medin & Waxman, 2010; Benelli, 1988; Gelman, 1989; Nelson, 1988). Quinn & Tanaka (2007) found that expertise within the same basic level category aided infants in the ability to discriminate other concrete level categories, emphasizing the role of expertise in category discrimination. Coley (2007) showed that children of eight years or older were more likely to categorize humans as being similar to primates and primates as being more similar to mammals than non-mammals, but younger children were not sensitive to these taxonomic groupings, further supporting the notion that intermediate level biological categories are learned rather than perceived. At least in humans, it appears that discriminations are not made solely on the basis of similarity of perceptual features. Categorizing stimuli according to biological taxonomies may then be presumed to be a uniquely human tendency.

Against this supposition, exciting recent work from neuroscience has demonstrated a common code for inferior temporal (IT) object representations in monkeys and humans (Kiani et al., 2007; Kriegeskorte, 2008). This work has demonstrated global representations along a continuum of inanimate to animate objects, but also within category distinctions between different taxonomic groups. Connolly et al. (2012) have suggested a continuum of activation representing categorical structure from insects to primates that mirrors the continuum from inanimate to animate objects. Connolly and colleagues suggest a categorical structure within the domain of animate objects that reflects the biological relations among species, suggesting that such categorization may be innate within primates. Furthermore, Murai and colleagues (Murai et al., 2004; Murai et al., 2005), using a familiarization-novelty preference task, have suggested that infant monkeys, chimpanzees and humans may spontaneously form categories at least at the global level, using categories of mammals, furniture and vehicles. The current experiments investigated the ability to make explicit classifications of more finite natural class distinctions in two other species of Great Ape; orangutans and gorillas. In the case of non-humans, categories would be based on shared observable features rather than on underlying knowledge of taxonomic class structures, particularly given that the only information provided involves visual features in two dimensional photographs. However, it was of interest to determine whether exemplars of more closely related groupings are more readily categorized together compared to more distantly related members of the same class. We predicted that orangutans may readily categorize stimuli from both concrete and intermediate level categories, whereas the gorilla might categorize stimuli more readily at the concrete level.

Subordinate categories, for example, ‘poodles’, are described as being concrete (or least abstract) along a hierarchy of abstraction because exemplars within such categories share many readily perceived features and can be easily distinguished from exemplars from other categories on a purely perceptual basis. Basic level categories, such as ‘dogs’, are described as being intermediate in terms of abstractness. The variability within an intermediate category is greater than the variability within a concrete level category. For instance, the more general category of dogs includes not only poodles but hounds, terriers, and many other types of dogs, which may vary in terms of size, color and distinctive features such as the long, short body of the dachshund. However, category distinctiveness is also increased such that members within an intermediate level category share fewer features with members of other intermediate level categories, whereas concrete exemplars may share many features with exemplars from other concrete level categories subsumed within the same intermediate category. For example, toy poodles may look a lot like the bichon frise but dogs may not look much like other mammals such as whales or marsupials. At the most abstract level, such as ‘animal’, superordinate category exemplars share even fewer perceptual features within a category but also share even fewer features with members of other abstract categories. The category ‘mammal’ may be considered superordinate to the basic level ‘dog’, but, in keeping with previously published studies (Roberts & Mazmanian, 1988; Vonk & MacDonald, 2002; Vonk & MacDonald, 2004; Vonk, Jett & Mosteller, 2012; Vonk et al., 2013), ‘mammals’ will be considered intermediate with ‘animal’ being considered the most abstract level category.

Of course, level of expertise determines category level to some degree. For an expert herpetologist, insects may be easily categorized at much finer levels than would be the case for a third grade science student, or a novice adult, for that matter. Thus, while the most generic level of categorization, e.g., insects may be a basic or intermediate level of category for the novice, a more highly specified category such as hymenoptera may serve as a basic level category for the herpetologist, with categories of wasps, bees, and ants–or even particular species of each, serving as subordinate or concrete levels. For the purpose of describing the current study, it is assumed that apes have no special expertise regarding unfamiliar members of the primate order, or broader class of mammals, birds, fish, insects, so the intermediate level will be used to refer to class while the concrete level will be used to refer to species discriminations.

Often abstract level categories can be discriminated only with some additional conceptual knowledge, rather than by relying on perceived shared attributes. For instance one would not know that insects, birds, amphibians, and mammals all belonged to the category ‘animal’ if one was unaware of their unobservable properties such as the ability to eat, breathe, reproduce, etc. The ease with which category membership can be determined on strictly a perceptual basis thus declines as one moves from concrete to abstract level categories. The terms concrete and subordinate, intermediate and basic, and abstract and superordinate have been used interchangeably. The former terms will be used exclusively throughout the remainder of this manuscript in keeping with previous research (Vonk & MacDonald, 2002; Vonk & MacDonald, 2004; Vonk, Jett & Mosteller, 2012; Vonk et al., 2013).

There is much evidence for concrete level discrimination learning in primates, (Fujita, 1987; Fujita & Matsuzawa, 1986; Fujita et al., 1997; Marsh & MacDonald, 2008; Yoshikubo, 1985) as well as in pigeons (Herrnstein, 1979; Herrnstein, Loveland & Cable, 1976), but there is much less evidence for intermediate or abstract level natural concept discrimination learning (although see Brooks et al. (2013) for evidence in rats). The difficulty with Brooks et al. (2013) is that all of the discriminations were also between animate and inanimate categories (e.g., chairs versus flowers and cars versus humans)–allowing the rats to make more global level discriminations. Recently, investigators (Autier-Dérian et al., 2013) have shown that domestic dogs are capable of categorizing many diverse dog species together into a single basic level “dog” category. The dogs were also able to discriminate between the species, demonstrating concrete level categorization. Caution is appropriate when making cross species comparisons as the reader must take into account possible differences in perceptual systems in different organisms. However, the species tested in relevant paradigms share sophisticated visual acuity and color vision making them suitable candidates for categorical discrimination research. In a more relevant study, Tanaka (2001) demonstrated that chimpanzees were able to discriminate exemplars both within intermediate level categories (at the concrete level) and between intermediate categories. The chimpanzees were trained to match exemplars that belonged to the same concrete level category and subsequently matched those exemplars to other members of the same intermediate, but different concrete level category, when a concrete level match was no longer an option. Because Tanaka’s chimpanzees were trained to make concrete level matches and then tested on intermediate level matches to the same stimuli it is difficult to compare the chimpanzees’ performance on the two levels of discrimination.

In one of only five published attempts to compare concept learning at different levels of abstraction in non-humans, Roberts & Mazmanian (1988) studied concept discrimination by pigeon, squirrel monkey and human subjects across three levels of abstraction, where abstraction was defined as the breadth of the category to be learned. At the concrete level, the subjects were asked to discriminate between photographs of one bird species (kingfishers) and other birds. At the intermediate level they were asked to discriminate between birds and other animals. Finally, at the most abstract level they were asked to discriminate between animals and non-animals. The authors found that humans easily discriminated concepts at all levels of abstraction, whereas both monkeys and pigeons had difficulty with the intermediate level discrimination. This finding was somewhat surprising because this intermediate level corresponds to the basic level that Rosch presumed was easiest for humans to learn (Keil, 1988; Mervis & Rosch, 1981; Rosch et al., 1976). Ross et al. (2003) suggest that children acquire such concepts by the age of six years. However, the results may confirm Mandler’s predictions (2000), particularly if the apes are performing conceptual rather than perceptual categorizations.

Vonk & MacDonald (2002) and Vonk & MacDonald (2004) trained orangutans and a gorilla to discriminate between photographs of members of their own species and humans, and between members of their own species and other primates (concrete level), between primates and other species (intermediate level) and between animals and non-animals (abstract level). The orangutans quickly learned the concrete and intermediate level discriminations and also learned the most abstract discrimination, but with slightly more difficulty. The gorilla subject appeared to have the most difficulty with the intermediate level discrimination, although she also learned all discriminations and showed significant positive transfer to novel stimuli. Thus both species of Great Ape demonstrated the ability to learn concepts at each level of abstraction, but the orangutans appeared to learn the basic or intermediate level discrimination the most readily while the gorilla learned it with the most difficulty. More recent follow-ups have shown chimpanzees to have the most difficulty with the most abstract discriminations (Vonk et al., 2013) in that they required more sessions to reach criterion as discriminations became more abstract, and were less likely to show significant transfer. In contrast, black bears showed significant transfer at all levels of concept discrimination, even when trained on the most abstract problems first (Vonk, Jett & Mosteller, 2012).

Some of the orangutans in the previous study (Vonk & MacDonald, 2004) learned to select photos of primates very rapidly (e.g., within three 10-trial sessions), despite the fact that the primates presented to them were of unfamiliar species. It was possible that they simply selected photos that they preferred. A spontaneous preference indicates that subjects may not have learned to abstract particular concepts or categories as being “correct” within the context of the experiment but implicitly preferred photos that happened to belong in the same taxonomic group. However, the very existence of a preference suggested that they did perceive primates as distinct from other species regardless of whether they explicitly recognized primates as belonging to a coherent category. Another possibility was that they selected photos of primates because these photos were more similar to photos that had previously been presented, although selecting these photos had not been reinforced. The latter explanation is unlikely, because of latent inhibition, the finding that it becomes more difficult to learn to respond to previously unreinforced stimuli due to retroactive interference. The fact that this was not the case for the orangutans suggests that they were attending to the category of the photo and not the specific exemplar itself. In addition, different and diverse primate species were presented in each subsequent transfer photo set, and transfer performance remained high. However, the question remained as to whether or not these apes would be equally likely to learn other intermediate level discriminations, including those more analogous to the bird/other animal discrimination tested by Roberts & Mazmanian (1988). Both chimpanzees and black bears were tested on an intermediate problem in which the two categories were equally inclusive and novel (carnivores for chimpanzees, or primates for bears, versus ungulates). The black bears performed better than the chimpanzees in terms of acquisition and transfer, suggesting the possibility of different mechanisms for forming the discriminations. Here, it was of interest to directly contrast the acquisition of matching intermediate level concepts to their acquisition of matching concrete level concepts in an identical procedure for the orangutans and gorilla tested previously.

The subjects were presented with a delayed matching-to-sample (DMTS) task in order to examine their understanding of several categories simultaneously. In contrast to the previous studies, where the subjects learned to discriminate a single category at a time, use of a DMTS procedure made it possible to have them categorize stimuli from several different species or classes within a single session. Each trial involved presentation of a sample that was replaced by two comparison photos once the subject attended to the sample. The reinforced comparison was a different photo of the same or different member of the same species or family as the sample (Experiment 1), or a photo of a different species from the same class as the sample (Experiment 2). The non-reinforced comparison was a photo of a member of a different primate species (Experiment 1), or a member of a different class (Experiment 2). Experiment 1 tested for concrete level discriminations in that the categorizations could be made by matching perceptual features of the stimuli. In Experiment 2 an effort was made to select photographs of species belonging to the same taxonomic class that were nonetheless perceptually quite distinct from one another. For instance, a photo of a stingray shared few features with a photo of the head of a blenny fish, but both belonged to the fish category. Backgrounds varied both within and between categories. Because the comparison stimuli did not share many perceptual features with the sample stimulus, the subjects’ ability to correctly match the photos might indicate the capacity for forming concepts at the intermediate level of abstraction.

Experiment 1

In Experiment 1, subjects were presented with photos of five different primate species (Japanese macaques, golden lion tamarins, proboscis monkeys) or families (gibbons, lemurs) and were rewarded for matching based on the species or family. This was a concrete level discrimination in which the exemplars within a category shared several physical features, such as color, and body shape. However, in two of the categories, (lemurs and gibbons) members of the same genus but different closely related species were included, and it was predicted that the subjects might have more difficulty when the match was a close relative compared to when the match was of the same species.

Materials and methods

Subjects

One female western lowland gorilla, Zuri (age 4), three male Sumatran orangutans, Dinding (44 years), Dinar (13 years), Molek (22 years) and one female Sumatran orangutan, Abby (42 years), participated in these experiments. All subjects were housed at the Toronto Zoo, Toronto, Ontario, Canada. The orangutan subjects were group-housed in an indoor exhibit and Zuri was housed separately at the time of testing, although she had auditory and visual access to the other gorillas at the zoo. The gorillas had access to both indoor and outdoor exhibits. Occasionally Zuri was integrated with the other gorillas for brief periods. Orangutans could view gibbons from their exhibit, as well as various bird, reptile, fish, and insect species. Dinding and Molek had been housed at the Yerkes primate research center many years prior and could see chimpanzees and bonobos from their enclosures there. Zuri could not view other primates from her enclosure but also had exposure to birds, reptiles, and insects. Four of the subjects had participated in one prior touch screen experiment (Vonk & MacDonald, 2002; Vonk & MacDonald, 2004), whereas Abby had participated in two prior touch-screen experiments (Vonk, 2003; Vonk & MacDonald, 2004). Only Abby had previously participated in a DMTS procedure (Vonk, 2003). Testing was approved by the Animal Care Review Board of York University, Canada under the direction of Suzanne MacDonald.

Materials

The photo set included 20 color photographs of 5 different primate species. There were 4 gibbon photos; two white-handed gibbons (Hylobates lar), one Mueller’s or Gray gibbon (Hylobates muelleri) and one dark-handed gibbon (Hylobates agilis). There were four lemur photos; three ring-tailed lemurs (Lemur catta), and one collared brown lemur (Eulemur fulvus collaris). Examples of these images appear in Fig. 1. There were also four photos each of Japanese macaques (Macaca fuscata fuscata), golden lion tamarins (Leontopithecus rosalia), and proboscis monkeys (Nasalis larvatus). Each species was depicted in various positions and orientations and in both close-up and far away shots. Some photos showed only the face of the subject whereas other photos showed the full body. Most photos depicted an individual whereas some included several individuals of the same species. The backgrounds of the photos varied both within and between categories. By varying such dimensions the extent to which irrelevant features might control responding in the task was minimized. A list and brief description of the photos used appears in Appendix S1. All photos were novel for all subjects.

Figure 1 Example images used in Exp. 1.

Example images from two categories in Exp. 2: gibbons (A–D) and lemurs (E–H).

Procedure

The experiment was programmed in Filemaker Pro 3 software for Macintosh. The photos were presented on a 13″ Apple touch screen monitor and were approximately 3″ by 4″ on the screen, separated by approximately 1.5″ and horizontally aligned. The monitor was placed against the bars of the subjects’ housing and they were required to either reach through the mesh holes to touch the screen (orangutans) or to reach underneath and touch the monitor (gorilla). In these experiments the experimenter sat behind the laptop, which was covered by a protective covering, connected to the touchscreen which was pushed right up against the subjects’ enclosure. The images on the touchscreen were mirror-reversed from the images on the laptop and the experimenter always gazed directly at the midpoint of the screen. The experimenter could not see the subject’s face or fingers, or the front of the touchscreen, when they made a response so could not react to the correctness or incorrectness of the choice until after the choice was made.

Subjects were tested individually at the same time each day. The orangutans received one to four sessions per day, two or three times a week, whereas the gorilla was given between five and ten sessions a day four days a week. The number of sessions was dependent on the availability of the subjects and their keepers. Each session consisted of ten trials. During a trial, a sample photograph was presented in the center of the monitor and stayed on screen until the subject attended to the photo and touched it, activating the touch screen. The two comparison photos subsequently appeared on the screen after a short delay (approximately 3 s). The subject was then required to select, by touching, only one of the two comparison photos. If he or she selected the photo that matched the species of the sample he or she was given a small highly preferred food reward by hand (M&Ms or dried fruits and nuts for the orangutans, and dried fruits or nuts for the gorilla) during presentation of a blank screen. If the subject made an incorrect response the screen advanced immediately to the blank screen and then to the next sample photo with no reward and no time-out. Thus intertrial intervals varied but were always less than one minute. Sessions continued until all ten trials were completed. Intersession intervals also varied but were always at least two minutes in length.

During each session, half of the photos were presented once and half of the photos were presented twice. If a photo had appeared as a correct comparison, it appeared as either a sample or as an incorrect comparison the next time it appeared within that session. This method discouraged a strategy of attending to individual exemplars and encouraged attending to the relationship between the sample and the comparison stimuli. The order of presentation and pairing of the photographs was randomized for each session. Thus each photo appeared as a sample, as a correct comparison or as an incorrect comparison during the course of the experiment. Each photo also appeared in both left and right positions on the screen across sessions. Within a session, half of the correct comparisons appeared on the left side of the screen, and half appeared on the right. Photos that appeared twice during some sessions appeared only once during other sessions.

Each species in the photos was represented in the sample twice and thus as a correct and incorrect match twice as well within each session. Thus gibbons, proboscis monkeys, tamarins, lemurs and Japanese macaques each appeared as samples on two of the ten trials within a session in random order. The photo used as a correct match was always different from the sample photo so that there were no identity matching trials. The same photos were used on each session.

For the first two sessions of the task, the subjects (except for Abby, due to previous training on a DMTS task) were given a small reward for simply touching the sample photo, as well as for making the correct choice. This was done so that the animals would learn to attend to, and select, the sample photo. After the second session, touching the sample photo was no longer reinforced. The experiment was considered complete when the subject was performing consistently, after a minimum of four blocks of five sessions (20 sessions or 200 trials). Subjects received four to six blocks of five sessions, depending on their level of performance. Abby completed the two experiments simultaneously as a control for the order of testing. She first completed five sessions of Experiment 2 and then five sessions of Experiment 1. From that point on, she completed one or two sessions of each experiment on each test day. The order of presentation of the two tasks was not counterbalanced for the other subjects because it was felt that presentation of the more visibly similar samples and exemplars in the concrete discrimination task would facilitate acquisition of the DMTS task that they had no prior experience with. Recall that these four subjects had not received any prior training on MTS procedures and none of the five had ever received identity-matching trials.

Results and discussion

As shown in Fig. 2 and confirmed by binomial tests, each subject performed at a level significantly greater than chance (50% correct) overall, all p’s < .001. Separate binomial tests for each subject also indicated how many sessions were required to reach levels of responding that were significantly above chance. Molek’s performance was significantly above chance by the second session, N = 20, p = .04. Dinding’s performance was significantly above chance by the fourth session, N = 40, p = .02. Abby and Zuri were performing significantly above chance by the sixth session, N = 60, p = .007 and .05 respectively. For Dinar, performance did not exceed chance levels until the 18th session, N = 180, p = .04.

Figure 2 Results from Exp. 1.

Average percent correct across blocks of 5 sessions (50 trials) for each subject in Experiment 1.

The fact that four of the five subjects reached above chance levels within the first six sessions of testing is impressive, particularly because the correct category differed on every trial. In addition, no training on the DMTS procedure occurred prior to the first session of testing. By contrast, subjects in similar experiments typically undergo extensive training with identity matching before being tested in conceptual matching procedures.

Recall that stimuli were randomly paired within each trial so that sometimes stimuli that had been paired on previous trials in prior sessions were re-paired and sometimes pairings were novel. It is possible that the subjects did not map the photographs on to concepts for each unique species but, rather, that they were rapidly able to learn associations between particular exemplars based on reward contingencies. In order to argue against this latter interpretation, paired t-tests were conducted to show that performance on novel stimulus pairings did not differ from performance on previous pairings, for the first six sessions, for any of the individual subjects, highest t2 = 1.67, p = .24. Only the first six sessions were considered because, after that, the likelihood of novel pairings decreased substantially. These data appear in Table 1.

Table 1 Performance in Exp. 1.

Percentage of correct responses in Experiment 1 on trials where exemplars comprised novel or prior pairings (Standard deviations in parentheses).

Subject	Novel pairings	Prior pairings	
Abby	77.5 (17.9)
N = 26	61.7 (21.9)
N = 34	
Dinar	58.2 (39.0)
N = 26	54.5 (13.8)
N = 34	
Dinding	66.2 (20.6)
N = 28	65.2 (16.9)
N = 32	
Molek	84.3 (15.0)
N = 23	78.3 (9.0)
N = 37	
Zuri	69.4 (30.3)
N = 28	64.9 (17.6)
N = 32	

In addition, one sample t-tests compared performance on both novel and prior pairings to chance (50%) for each subject for the first six sessions. Only Molek was above chance on both novel (M = 84%, SD = 15%, t4 = 5.09, p = .007), and prior pairings (M = 78%, SD = 9%, t4 = 6.78, p = .002). Abby was above chance on only novel pairings (M = 78%, SD = 18%, t4 = 3.07, p = .05). The other three subjects were not above chance on either novel or prior pairings within the first six sessions. These results lend no support to the idea that subjects relied upon associations that they formed during the course of the experiment between particular stimuli pairings and reward in performing this task.

In order to determine if subjects were differentially accurate at matching photos depending on the species or family of the animal depicted, separate univariate ANOVAs of the subjects’ scores on each session, with sample species (species of the animal in the sample stimulus) and incorrect species (species of the animal in the non-reinforced comparison stimulus) as independent variables were conducted for each subject. ANOVA assumes normality of data and homogeneity of variance–conditions that were satisfied by the current data. Table 2 displays the average scores for each subject for each type of discrimination, according to which specie was depicted in the sample. Dinding was influenced by both the species of the sample stimulus, F4,25 = 5.66, p = .002, and of the non-reinforced comparison, F4,25 = 4.23, p = .01. He scored above 66% for all species except when the sample was a Japanese Macaque; on those trials he was below chance. He also tended to score near chance levels when the non-reinforced comparison was a proboscis monkey or a golden lion tamarin, indicating that he preferred to select those images.

Table 2 Performance by category.

Average percent correct for each subject across each type of discrimination (according to the image depicted in the sample) for Experiment 1. Standard deviations in parentheses.

	Zuri	Molek	Dinar	Dinding	Abby	
Tamarin	79 (03)	73 (13)	84 (10)	78 (12)	75 (13)	
Gibbon	78 (10)	84 (09)	52 (12)	69 (15)	63 (10)	
Jap. Mac.	81 (08)	76 (21)	58 (13)	42 (13)	75 (10)	
Lemur	71 (14)	74 (09)	46 (10)	70 (13)	70 (20)	
Proboscis	36 (13)	60 (07)	61 (14)	66 (18)	73 (13)	

Dinar’s choices were influenced by the sample stimulus, F4,25 = 8.7, p = .001. He matched accurately when the sample photo was of a golden lion tamarin but matched inaccurately with every other type of sample. He was apparently distracted by the non-reinforced comparisons, F4,25 = 8.78, p = .001. He made many errors when the lion tamarin was the incorrect choice, indicating a preference for photos of tamarins regardless of whether they were the correct matches or not. He also tended to select photos of gibbons and lemurs when they were not correct choices. It is not unexpected that he might prefer to choose gibbons over other primates because gibbons were visible from the orangutan exhibit and were the only other primate species, other than humans or orangutans, that the orangutans were familiar with. Dinar’s results suggested that he did not use a generalized concept to perform the task, but are not inconsistent with an ability to discriminate the different species of primates.

Molek’s choices were affected by the non-reinforced comparison, F4,20 = 5.63, p = .003. He performed accurately on all trials except when the non-reinforced comparison depicted a Japanese macaque, indicating that he selected the Japanese macaque photos often when they were not correct.

Abby’s choices were not significantly affected by the sample species or by the non-reinforced species, although the effect of non-reinforced species approached significance, F4,20 = 2.64, p = .08.

Zuri’s choices were influenced by the sample species, F4,15 = 13.06, p = .001. She scored above 70% on all trials except for when the sample photos were of proboscis monkeys. She was also affected by the species depicted by the non-reinforced comparison photos, F4,15 = 3.83, p = .02. She chose accurately on trials when proboscis monkeys were depicted in the incorrect comparisons, consistent with the idea that she simply avoided photos of proboscis monkeys regardless of whether they were correct matches or not. She also did well when tamarins were the non-reinforced comparisons but not as well when gibbons, Japanese macaques or lemurs were the non-reinforced comparisons.

None of the subjects, with the exception of Dinar, appeared to have significantly greater difficulty with the gibbon or lemur species, despite the fact that these photo sets included different species within the same family. This finding implies that the greater perceptual difference between the sample stimulus and the comparison stimulus did not influence responding for most of the subjects.

The results indicated, for the most part, that the subjects matched above chance at this concrete level. They sometimes displayed idiosyncratic preferences for photos of particular species but, overall, they discriminated between members of various primate species, consistent with earlier research in which they selected photos of orangutans or gorillas and avoided selecting photos of other primates (Vonk & MacDonald, 2002; Vonk & MacDonald, 2004). The present experiment extends this finding to photographs of unfamiliar species, and to a new procedure. Use of a DMTS task has an advantage over the two-choice procedure in that no single exemplar is associated with reward or non-reward. Instead it is the relationship between, or the pairing of, stimuli that determines the reward contingency. The fact that the subjects were no less accurate with novel than prior pairings is thus an important finding and argues against simple stimulus reward associations. Other experiments with the same subjects revealed that they do not learn all DMTS tasks equally rapidly (Vonk, 2002; Vonk, 2003), implying that there is something special about the nature of these discriminations making the categories readily perceivable by these species.

The subjects differed from each other in terms of which species they preferred to select and which species they had the most difficulty matching. In general, however, they seemed to have difficulty with the proboscis monkey photos, which was unexpected because those photos were mostly, although not exclusively, of the same animal from various positions but with the identical background, whereas photos of the other species varied along more dimensions and sometimes even depicted different species. This result therefore argues against a reliance on perceptual similarities to make the discriminations or attention to irrelevant background details. The difficulty may have stemmed from the fact that the proboscis monkey subjects tended to fill a smaller percentage of the entire photo.

Above chance levels of performance were obtained within the first six sessions by all but one of the subjects, indicating that accurate responding did not require extended training or learning of associations between the exemplars. Performance was no more accurate on trials where specific exemplars had been paired before than on those trials on which stimulus pairings were novel. However, Dinar’s pattern of responding suggested that he often simply selected photos that he preferred. Concrete level discriminations can be made on a perceptual basis and are not necessarily demonstrative of abstract concepts representing the species depicted. Thus performance may be expected to decline in Experiment 2 where photos had to be matched according to intermediate level categories. Exemplars within these categories shared fewer physical features.

Experiment 2

The second experiment was of critical interest. Few researchers have examined whether non-human primates spontaneously classify other species according to biological or taxonomic categories (see Brown & Boysen (2000) for one example). Could these subjects match photos of members of various taxonomic classes (such as birds, reptiles, insects, mammals, and fish) despite the fact that exemplars within a category would share only some features, and may also share features with exemplars from another category? This experiment tested their abilities to form concepts at an intermediate level of abstraction, and might corroborate previous findings (Vonk & MacDonald, 2002; Vonk & MacDonald, 2004).

Method

Subjects

The subjects were the same five animals that participated in Experiment 1.

Materials

The photo set included thirty novel color photos, six from each of the following taxonomic class categories: birds, insects, mammals, fish and reptiles. Photos included single or several individuals, pictured close-up or at a distance, faces or entire bodies. The subjects in the photos were also pictured in a variety of orientations and postures. Figure 3 depicts sample images from the categories of birds and reptiles. Each photo appeared once during each session. Whether or not the photo appeared as a sample, correct, or incorrect choice was randomly determined on each session. Photos were also randomly paired. Within each session one different exemplar from each taxonomic group appeared as a sample twice and as an incorrect choice twice as well. Thus, out of ten trials within a session, two of the sample photos were fish; two were birds, insects, mammals and reptiles. The same thirty photos were used in each session. A list and description of the photographs appears in Appendix S2.

Figure 3 Example images from Exp. 2.

Examples of images from two categories used in Exp. 2: Reptiles (A–F) and Birds (G–L).

Procedure

The procedure was the same as in Experiment 1, except for the different materials noted above. In addition, the subjects did not have to be rewarded for touching the sample photo on the first two sessions, because they had already mastered the DMTS procedure. Each subject received four or five blocks of five sessions depending on the number of sessions required to reach a stable level of responding.

Results and discussion

As shown in Fig. 4, and confirmed with binomial tests, each subject performed significantly above chance (50%) overall, N = 200 or 250, all p’s = .001. Individual binomial tests were conducted to determine how many sessions were required for each subject to achieve above chance levels of performance. Molek and Dinar required only two sessions to perform above chance, N = 20, both p’s = .04. Abby performed above chance after only four sessions, N = 40, p = .04, and Dinding performed above chance after only six sessions, N = 60, p = .05. Zuri, on the other hand, required 14 sessions to reach above chance levels of performance, N = 140, p = .04.

Figure 4 Results from Exp. 2.

Average percent correct across blocks of 5 sessions (50 trials) for each subject in Experiment 2.

In order to argue against an interpretation favoring rapid learning of associations between particular stimuli, paired t-tests were conducted to show that performance on novel stimulus pairings did not differ from that on previous pairings, for the first six sessions, for any of the subjects. These data appear in Table 3. Performance did differ between novel and prior pairings but only the difference for Dinar reached significance, and his performance was actually better for novel (M = 74%, SD = 5%) than for prior pairings (M = 52%, SD = 11%), t4 = 3.31, p = .05). Again, there was no evidence that subjects were learning to choose correctly based on remembering associations between the stimuli and patterns of reward.

Table 3 Performance in Exp. 2.

Percentage of correct responses in Experiment 2 on trials where exemplars comprised novel or prior pairings (Standard deviations in parentheses).

Subject	Novel pairings	Prior pairings	
Abby	49.2 (20.9)
N = 39	81.9 (26.1)
N = 21	
Dinar	74.5 (5.2)
N = 43	51.8 (11.2)
N = 17	
Dinding	58.8 (16.3)
N = 41	78.4 (21.7)
N = 19	
Molek	77.9 (7.4)
N = 41	71.6 (29.9)
N = 19	
Zuri	51.3 (7.4)
N = 44	51.0 (47.5)
N = 16	

In addition, one sample t-tests compared performance on both novel and prior pairings to chance (50%) for each subject for the first six sessions. Performance was above chance on only novel pairings for Molek, (t5 = 9.62, p < .001) and Dinar, (t5 = 11.44, p < .001). Performance on prior pairings alone was significantly above chance for Abby (t4 = 2.73, p = .05) and Dinding (t4 = 2.92, p = .04). Zuri’s performance was not above chance for either novel or prior pairings within the first six sessions, highest t5 = 0.44, p = .68.

Individual univariate ANOVAs of the subjects’ scores with class (of the sample photo) as the independent variable, for each subject, revealed an effect of class that was significant for Dinar alone, F4,20 = 6.82, p < .001. Dinar scored above 69% correct on all discriminations except for birds, on which he performed close to chance. Dinding also had difficulty with bird trials as well as with mammals, F4,20 = 2.40, p = .08. The class of the animals depicted in the sample did not significantly affect the performance of Molek, (F4,14 = 1.54), Abby, (F4,14 = 2.06), or Zuri, (F4,14 = 1.01) all p’s > .05. Average percent correct on trials with each type of discrimination are displayed in Table 4.

Table 4 Performance by category.

Average percent correct for each subject across each type of discrimination (according to the image depicted in the sample) for Experiment 2. Standard deviations in parentheses.

	Zuri	Molek	Dinar	Dinding	Abby	
Bird	65 (13)	75 (17)	53 (06)	57 (06)	80 (08)	
Fish	63 (05)	85 (06)	73 (08)	69 (16)	68 (10)	
Insect	60 (14)	65 (06)	71 (05)	68 (04)	70 (14)	
Mammal	80 (27)	85 (10)	72 (05)	58 (08)	73 (10)	
Reptile	60 (16)	73 (22)	70 (11)	72 (11)	60 (08)	

Only one of the subjects, Dinar, showed significant differences in accuracy based on which class the sample photo belonged to. Dinar may have been distracted by preferences for particular photos in both experiments. However his accuracy in this experiment was higher and his performance was more consistent. The other subjects were not significantly distracted by preferences for photos of animals belonging to particular taxonomic categories.

The orangutans were able to rapidly discriminate amongst species from different taxonomic classes and did not have to learn to make associations between the exemplars. However, the gorilla did not reach significant levels of performance until the third block of sessions suggesting that she was not as predisposed to making these discriminations initially. In addition, her accuracy tended to be better on Experiment 1, whereas the reverse was true for at least two of the orangutan subjects, Molek and Dinar, (although these tendencies were statistically significant only for Dinar when performance was compared across experiments, (repeated measures ANOVA, F1,24 = 8.18, p < .009). The increase in performance for these two orangutans might be attributed to prior experience with the DMTS procedure in Experiment 1. However the same increase was not found for other similarly trained subjects (Zuri and Dinding). In addition, Abby, who had been previously trained on DMTS tasks in a different study (Vonk, 2003), and who was tested on both experiments simultaneously, performed equally well across experiments. This finding is not consistent with the claim that performance on the more concrete level task might be superior if task ordering was not a factor.

Conclusions

It appears that orangutans learn intermediate level discriminations at least as readily as they learn concrete level discriminations (see also Vonk & MacDonald, 2004). This is consistent with the prediction made for humans (Keil, 1988; Rosch et al., 1976), and possibly with findings from chimpanzees (Tanaka, 2001, although see Vonk et al., 2013), but not from squirrel monkeys, pigeons (Roberts & Mazmanian, 1988) or gorillas (Vonk & MacDonald, 2002). The gorilla subject in the present experiments seemed to learn the concrete level discrimination more rapidly despite the fact that she was simultaneously learning the DMTS task for the first time. However, it is difficult to make cross-species comparisons when fewer than five subjects of each species is tested, particularly given the extreme individual subject differences in these experiments and others. Research with young humans suggests that the intermediate or basic level concepts are learned before subordinate or concrete level discriminations (Rosch et al., 1976). A general assumption is that categories acquired earlier in ontogeny are also more accessible to more phylogenetically removed species. Although the current study was not designed to address developmental changes in concept acquisition, the fact that at least one gorilla showed greater facility with concrete level categories is interesting and is worthy of further exploration.

Intermediate level categories are those that maximize within-category similarity and distinctiveness relative to between-category similarity (Medin & Smith, 1984; Mervis & Rosch, 1981). It is unclear whether these categorizations are readily made because of an inherent tendency to detect perceptual similarities within classes and dissimilarities between classes, or whether such categorizations must be learned through experience with the exemplars or natural instances (Medin & Smith, 1984). It is also unclear whether experiments such as these direct the creation of a concept, or merely provide evidence for pre-existing concepts in the subjects (Huber, 1999; Roitblat & Von Fersen, 1992). However, the fact that subjects achieved above chance levels of responding with both of these discriminations more rapidly than they did with different kinds of MTS tasks (Vonk, 2002; Vonk, 2003), despite learning the DMTS procedure for the first time here, indicates that the categories may have been spontaneously perceived, or at the very least, relatively easy for these subjects to acquire. The current experiments provide the most direct comparison between acquisition of various concrete and intermediate level category discriminations made at a perceptual level in the absence of additional information, such as labels or biological facts. They therefore provide evidence that categories at various levels of abstraction can be acquired in the absence of linguistic labels or biological knowledge, in at least two of our closest primate relatives.

Previous studies have shown that items may be correctly classified spontaneously and without regard to experimental training but have not directly contrasted classification at various levels of abstraction (Murai et al., 2004; Murai et al., 2005). Cerella (1979) investigated the ability of pigeons to discriminate oak leaves from leaves of other species. Results from his series of experiments lent support to the idea that these taxonomic classifications were made spontaneously and did not involve induction. Significant transfer was made to multiple unique positive exemplars after training with a single exemplar, and transfer did not depend upon contrasting the original instance to negative instances. Typically, learning of experimentally defined categories requires presentation of both positive and negative stimuli (Sutton & Roberts, 2002) and is improved by the presentation of multiple exemplars (Katz, Wright & Bachevalier, 2002; Sutton & Roberts, 2002; Wright & Katz, 2007). That Cerella’s pigeons did not require this experience might suggest a pre-existing concept for oak leaves. The number of exemplars presented was also limited in the current experiments.

Furthermore, Cerella’s pigeons had difficulty discriminating between instances of oak leaves that varied on specific dimensions, despite the fact that they could discriminate between leaves of different species that sometimes shared features. These findings also suggest that pigeons may be hardwired to perceive oak leaves as belonging to the same species, unique from other species of leaves, as opposed to learning the discriminations by attending to distinct physical features in the stimuli. In the current study, at least three orangutans learned rapidly to judge animal members of different taxonomic classes as belonging to the same category, despite lacking experience with many of the species depicted in the test stimuli. The rapid learning could not be attributed to learning associations between specific exemplars because both the orangutans and the gorilla were at least as accurate with newly paired as with previously paired exemplars. Thus it is possible that concepts for intermediate level categories are readily extracted from shared perceptual information between the stimuli. The gorilla subject appeared to learn to classify the stimuli similarly after several sessions, suggesting that she was capable of perceiving distinctions between the categories tested but did not do so immediately.

In an interesting test of spontaneous classification, Brown & Boysen (2000) presented chimpanzees with pairs of photos depicting different species of animals and required them to identify the pairs as being either the same or different. The chimpanzees showed above chance categorization despite not being differentially reinforced for their responses. Interestingly, whereas they were slightly more likely to classify tigers and housecats as the “same”, they were not more likely to classify gorillas and chimpanzees as the “same”, relative to the other comparisons tested. Therefore, whereas there was some evidence for more intermediate level discriminations, in general, the spontaneous discriminations made by the chimpanzees were more analogous to concrete level discriminations. Findings from this study are limited based on the fact that chimpanzees were presented with specific pairs and might indicate them to be different but could not also indicate that they were more similar to each other than to other potential pictures. For example, they may have indicated that chimpanzees were different from gorillas but still might have had the capacity to categorize chimpanzees and gorillas as more alike than chimpanzees and lions. This capacity was not tested, however. In one of only two other studies to investigate various levels of abstraction in chimpanzee concept formation (Tanaka, 2001) the procedure does not allow comparison of the relative ease with which the two levels of concept discrimination were achieved. In Vonk et al. (2013), only two chimpanzees were tested in different orders as an attempt to control for order effects in learning the discriminations at various levels. One chimpanzee did not show evidence of concept acquisition at any level, and the other found discriminations more difficult as they became more abstract. The current study was able to test for multiple possible category matches in the same session without setting one category as correct or incorrect as in Vonk et al. (2013). In the current study, we were also able to assess whether matches were more difficult for particular species or class comparisons and whether errors revealed associative learning, perceptual confusions or untrained preferences.

In Brown and Boysen’s study (2000), chimpanzees were less likely to judge two different chimpanzees as being the same, relative to their judgments for cats, tigers, fish and gorillas. This result is not surprising, given that animals may be more inclined to detect individual differences between members of their own species (Martin-Malivel & Okada, 2007). Face processing in particular may be specific to one’s own species in other primates (Dufour, Pascalis & Petit, 2006). In the current experiments, orangutans were slightly less likely to correctly match members of the same primate species, as compared to their performance for members of broader taxonomic groups. Perhaps this finding is due to a greater tendency to perceive individual differences amongst other primate members. It is possible that basic or intermediate level discriminations are most critical to an animal’s survival and that, at the more concrete level, even finer perceptual discriminations are made.

The results of the current experiments cannot rule out a perceptual basis for making taxonomic classifications, because a certain degree of perceptual overlap is necessarily evident among members of a class. However sufficiently diverse stimuli were used here, to rule out the use of single or few features. The subjects were required to make judgments based on relatively few exemplars and reached high levels of performance after few sessions. This result may be considered more consistent with the idea of an innate mechanism for distinguishing among members of a class on a perceptual basis (Cerella, 1979), as opposed to inducing rules about the associations between stimuli (Gelman, 1989; Mandler, 2000). However, subjects who did not initially perform at high levels did learn to match at very high levels. This finding suggests that even when categorizations are not made spontaneously, other non-human Great Ape species do possess the capability for applying rules based on the presence or absence of multiple relevant features.

Hampton (1998) demonstrated the influence of biological knowledge on humans’ classifications of similar natural stimuli; birds, fish, insects and animals. This knowledge influenced classification more than it influenced typicality judgements. The present results are not consistent with the idea that language (Benelli, 1988; Nelson, 1988) or scientific knowledge (Inagaki, 1989) is necessary for making natural taxonomic classifications–at least not at the perceptual level of categorization. Instead it would appear that our ability to categorize organisms based on biological similarities is shared with other members of the animal kingdom, at least with other non-human primates. These distinctions can clearly be made on the basis of presentation of two-dimensional stimuli depicting species with which the subjects have had no experience.

In accord with the hypothesis that language is not needed to support the categorization of natural stimuli, human infants have been shown to categorize in a manner that corresponds to those categorizations made by adults on the basis of biological knowledge (Eimas & Quinn, 1994). Clearly infants do not yet hold conceptual representations of biological categories and yet, the fact that these perceptually based discriminations correspond to mature adult conceptual discriminations is important. As concepts are learned they may be influenced by cultural contexts, particularly the labels provided by natural languages (Davidoff, 2001; Whorf, 1956); However, Rhodes & Gelman (2009) have shown that cultural context is important for the categorization of artifacts, but less so for natural objects. Following a meta-analysis of concept studies across cultures, Malt (1995) concludes that there is significant structure provided by natural categories to evoke similar categorization in different cultures that vary with regard to familiarity with the objects (e.g., bird species). Across cultures, there was great convergence between folk concepts and scientific concepts. However, she concludes that the issue is more complex in that culturally specific beliefs may influence categorization at more abstract levels. Data from such cross cultural studies and comparative studies (such as this one) converge to suggest that the ability to designate linguistic labels for categories is not necessary for creating at least perceptual representations of biological categories (see also Wasserman & Devolder, 1993). Perhaps both infants and non-human primates have acquired the ability to perceive categories, an ability that may underlie the capacity to develop abstract concepts whose full emergence may yet depend upon the development of language. This latter conjecture has yet to be proven. The current experiments are a start in that direction.

In defining “concepts” it is important to distinguish between perceptual versus conceptual processes (Eimas & Quinn, 1994; Huber, 1999; Mandler, 2000; Premack, 1983). Because the subjects in the current study were able to correctly match members of various taxonomic groups after a brief delay between sample and test stimuli, it is possible that they maintained a representation of the category itself and not only of the particular sample exemplar. This possibility is made more likely by the important finding that the subjects did not rely upon remembering previous pairings of stimuli. In addition the subjects here were required to make an instrumental response to the stimuli. These results might thus constitute evidence for a conceptual versus a purely perceptual representation. Use of a concept is implicated when an individual is able to form a coherent category from exemplars displaying some shared and yet many distinctive features (Spalding & Ross, 2000). Analysis of single features in isolation is not sufficient for the formation of abstract concepts. Instead, the individual must combine and compare various features and make some determination as to which features are deemed critical for category membership. In the current experiment, orangutans and one gorilla were able to analyze various perceptual aspects of two-dimensional photographs and to use this information to represent distinct categories. Further work is needed to clearly discern the extent to which such categories are perceptual versus conceptual in nature.

Supplemental Information

Supplemental Information 1 Stimuli list

Click here for additional data file.

The cooperation and support of the staff at the Toronto Zoo was greatly appreciated. Special thanks to Dianne Devison, Bev Carter, Vanessa Phelan, Gerri Mintha, Charles Guthrie, Tim McCaskie, Connie Wiederman, Heidi Minicke, Michelle Smith, Karyn Tunwell, Bridget Burke-Johnson, Andrea Beatson, Rick Vos, Mark Bongelli, David Partington, Des Macguire, John Armstrong, and Jackie Craig, without whose assistance this study would not have been possible. I would like to thank Suzanne MacDonald and Robert Sorge for comments on an earlier version of this paper. I also thank Jon Toth for providing protective equipment. Suzanne MacDonald supported the research as an advisor/mentor.

Additional Information and Declarations

Competing Interests

Author Contributions

Animal Ethics

The author declare there are no competing interests.

Jennifer Vonk conceived and designed the experiments, performed the experiments, analyzed the data, contributed reagents/materials/analysis tools, wrote the paper.

The following information was supplied relating to ethical approvals (i.e., approving body and any reference numbers):

The Animal Care Review Board of York University, Toronto, Canada provided a blanket approval for zoo research under Suzanne E. MacDonald’s direction. Studies off site are considered field research and not given full review so do not receive permit numbers.

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
