# Peer review of "Matching based on biological categories in Orangutans (Pongo abelii) and a Gorilla (Gorilla gorilla gorilla)"

_PeerJ, doi:10.7717/peerj.158_

## Round 0.1 · original submission · Minor Revisions

Please consider all points raised by the reviewers and address them by making appropriate changes to your manuscript.

·

Basic reporting

Overall, an interesting paper and generally well written, this study will be of interest to behavioural scientists. I think the paper is publishable, subject to some revisions.
First, at a general level, animals' discriminative skills should be discussed with some reference to perceptual differences. The subjects (great apes) were tested using visual discriminations, based on photos, which is no doubt appropriate for these species. However, the author also makes reference to discriminative skills in other species (eg dogs) whose skills may well have different perceptual bases. The paper should acknowledge that the tests used were specifically testing visual skills, and contextualise discussion accordingly.
Second, there should be more discussion of generalisation from the relatively small number of animals used. I am well aware of the difficulties of doing studies with zoo animals - it almost always involves a small N. But here, I felt the paper would benefit from considering how general the demonstrated abilities are (especially for gorillas).
More specific points: there should be more detail of how the apes were kept. This is particularly relevant regarding proximity of other species, and the subjects' experience of other primate species. Following on from this, I am sceptical of the implication that specific skills "pre-existed" (p. 24) - ie are hard-wired - unless we know much more about these individuals' histories.
Page 8, paragraph 2 - this was not entirely clear regarding chimps. Please clarify what "most difficulty with the most abstract distinctions" means.
Page 9, lines 4-5 - needs clarification.
P.12. Please explain why different numbers of trials were used.
p.26. Discussion of how humans classify other species should acknowledge cross-cultural differences in classification.
I found the tables and figures rather confusing. In table two, the caption says "percentages", yet the numbers are all less than 1. The figures do not seem to show any significant improvement over time - please add more information.

Experimental design

The overall design is satisfactory, although I would encourage the author to include more detail of statistics. In particular, the justification of the ANOVA, and the assumptions made in its use, should be clarified (c. page 15)

Validity of the findings

See comments above.

Reviewer 2 ·

Basic reporting

(referring to introduction): excellent.

see general comments also

Experimental design

Thorough and circumspect and avoiding many potential pitfalls. The researchers took every possible precaution to avoid preferences and ambiguous interpretations of results. Because of the design the experimenter was able to determine that subjects did not rely upon associations they formed during the course of the experiment between particular stimuli pairings and reward in performing this task.

Validity of the findings

The experiments are carefully controlled, the test variables rigorously controlled and the statistical tests appropriate. The results therefore seem robust and the findings very convincing. We are also told the prehistory of experiences of these specific great apes taking part in these experiments reassuring one that these were novel experiences and that it was not prior learning or extensive testing which had produced the current results.

Additional comments

This is an exciting paper concerned with a possible distinction recognized and known in humans between specific and concrete and global categorizations of objects, object classes and overall classifications without the benefit of language. They used biological classifications of species, family and class. Four orangutans and one gorilla were presented with delayed matching-to-sample tasks and, in two experiments, taken through levels of perceptual matching to abstract matching., i.e. same species group or different families and, finally, fitting individual images into appropriate classes of insect, reptiles, birds or mammals respectively. Impressively, the paper found (for the first experiment) that five of the apes performed above chance levels within the first six sessions. For the second experiment testing the apes’ ability to assign specific photos to classes most apes performed again well above chance, suggesting either that they spontaneously recognised the categories or were able to learn about such categories with relative ease and speed. The authors were cautious and parsimonious in their interpretation and, despite their strong results, resisted the temptation to attribute abstract thinking to the apes but they are saying that biological categorisations do not require specialised taxonomy knowledge but, in this case, a recognition of the basic rules governing the classifications. Their experiments have opened the debate for further studies in this field on how categorisation occurs in pre-linguistic contexts.
I have very little to add in a critical sense and highly recommend publishing.
There are a few minor grammar and other errors in the paper that the author should correct before final submission. One glaring example was the formulation that traits are ‘shared in common’ –a bad logical error/doubling up: it’s either ‘have in common’ or ‘share xyz with someone’…(leaving out ‘in common’) because when one has certain traits in common with someone else they are, in fact, shared traits (the male stallion problem).
There are other small (i.e. readily fixable) issues in presentation: The discussion/conclusion is not as good and tight as the introduction and I would hope that the author could strengthen the discussion by referring back to the introduction in which the categories and distinctions were very well presented indeed and theoretically cogent. Some of the sparkle is gone in the discussion and that is a pity and could be improved simply by using and expanding upon the same sharply defined theoretical concepts and issues as in the beginning of the paper.

Reviewer 3 ·

Basic reporting

No comments

Experimental design

No comments

Validity of the findings

No comments

Additional comments

This paper investigates the conceptual organization of biological stimuli in perceptual and conceptual categories by non-human primates (gorillas and orangoutangs). In the human literature there has been a debate on the role of language vs. perceptual similarity for the formation of categories, in relation to the level of abstraction implied by a given category. The present study aims at investigating this topic from a comparative perspective, testing non-verbal animals that are phylogenetically close to humans. Categorizing stimuli according to biological taxonomies has hypothesized to be a uniquely human tendency. Contrary to that claim, recent neuroimaging and behavioral data suggested that non-human primates may spontaneously form categories of biological objects in a similar way to what humans do.

The current paper investigated the ability to make explicit classifications of natural class distinctions, to determine whether exemplars of more closely related groupings are more readily categorized together compared to more distantly related members of the same class.

In the present study, overall 5 subjects were required to match images based on biological classifications at the level of species, family or class.

The results of the first experiment indicate that the subjects can form categories at the concrete (species) level, even when confronted with images of unfamiliar primate species. This task however can be solved on the basis of perceptual similarity and does not imply sophisticate abstraction abilities. In Experiment 2 subjects proved able to solve the task also based on a more abstract level of categorization, i.e. using the intermediate categories of taxonomic classes. Moreover, at least orangoutangs proved able to learn intermediate-level categories as readily (or even more readily) as concrete categories, in line with the human developmental literature. On the contrary the gorillas seemed to find it easier to form categories at the concrete level. However, the small sample size prevents from making meaningful comparisons between the performance of the two species (orangoutangs and gorillas).


Unfortunately the present study cannot provide any information on the role of experience with different living creatures in the relative facilitation for forming intermediate-level categories displayed by some of the subjects (for example see page 23, first paragraph). This most interesting issue can be assessed only by investigating the performance of subjects reared in strictly controlled environment and/or tested at a very early age after birth. Authors should be careful in their claims on this point (e.g., at page 24).

The introduction contains some repeated information, which should be avoided (e.g. the work of Autier-Dérian et al., 2013 is described two times for no apparent reason). Moreover, the structure of the introduction itself is somehow unclear, some topics seem to assed repeatedly in different parts of the text. It would be helpful to make the introduction more dense and to provide a clearer structure for the reader. It could be also helpful to add, at the end of the introduction, a short paragraph highlighting the main conclusions and open issues originated from the existing literature.

Similar problems are present also in the general discussion. The most problematic issue with the present paper is that it is not clear which are the conclusions that can be drawn from the results obtained. It is necessary to highlight the open questions that the study wanted to asses, and how (if) the results obtained provide novel information to answer these questions.


Minor comments

In the discussion session evidence is reported that categorization of own species images tends to occur at a more concrete level than categorization of distantly related species (page 24 and 25). This is likely to be due to the special status of own-species, which could be perceived as a category per se and elicit more detailed elaboration. It could be also appropriate to discuss this evidence in relation to the "other species" effect observed during face perception in humans.


The abstract should mention which are the conclusions obtained from the present study.


It would be very helpful to include images of the stimuli, since their perceptual appearance is crucial to the interpretation of the results.

---

## Round 0.2 · Minor Revisions

The following points still need to be addressed.
1. The following sentence in the Introduction is not entirely clear to me. “In her view, basic level categories may be categorized perceptually before superordinate categories, but with regards to conceptual categories, which are based on kind as opposed to perceptual similarity, more global, abstract categories such as animals, foods etc. may emerge first.”. Please can you improve it?
2. In the new sentence “We predicted that orangutans may readily category stimuli from both concrete and intermediate level categories, whereas the gorilla might categorize stimuli more readily at the concrete level.” Correct category to categorize.
2. You use the word ‘hoofstock’ in the Introduction but it is not exactly clear what is meant by this.
3. The first sentence of the Results and Discussion section of Experiment 1 states “As shown in Figure 2 and confirmed by binomial tests, each subject performed significantly greater than chance..”. Please insert ‘at a level’ after ‘performed’.
4. The second sentence under the heading Experiment 2 doubles up on commas and brackets - ,( and ,). Please choose which of these you want to use.
4.One of the reviewers said, “I found the tables and figures rather confusing. In table two, the caption says "percentages", yet the numbers are all less than 1. And you replied, “A: The numbers are less than one because percentage correct was calculated as a score between 0 and 1 (e.g. 55% = .55).”. Tables 2 and 4 remain a problem because they are not percent scores. Please change to the 0 to 100% range.
5. In response to a reviewer you changed the following sentence: “Although the current study was not designed to address developmental changes in concept acquisition, the fact that at least one gorilla showed greater facility with concrete level categories is interesting and should be explored further.” Replace ‘should be explored’ by ‘worth exploring’.

---

## Round 0.3 · accepted · Accept

Thank you for making the final corrections.